# ESCAPING STOCHASTIC TRAPS WITH ALEATORIC MAPPING AGENTS

## ABSTRACT

When extrinsic rewards are sparse, artificial agents struggle to explore an environment. Curiosity, implemented as an intrinsic reward for prediction errors, can improve exploration but fails when faced with action-dependent noise sources. We present aleatoric mapping agents (AMAs), a neuroscience inspired novel form of curiosity modeled on the cholinergic system of the mammalian brain. AMAs aim to explicitly ascertain when dynamics of the environment are unpredictable, even if those dynamics are induced by the actions of the agent. This is achieved by generating separate forward predictions for the mean and aleatoric uncertainty of future states, reducing intrinsic rewards for those transitions that are unpredictable. We demonstrate that in a range of environments AMAs are able to circumvent action-dependent stochastic traps that immobilise conventional curiosity driven agents.

## 1 INTRODUCTION

Efficient exploration is a central problem in reinforcement learning. Exploration is particular challenging in environments with sparse rewards—requiring agents to navigate with limited guidance (e.g. Sutton & Barto (2018); Pathak et al. (2017); Burda et al. (2018b), see Weng (2020) for a review). A notable exploration method that effectively deals with sparse rewards is curiosity driven learning—where agents are equipped with a self-supervised forward prediction model that employs prediction errors as intrinsic rewards Schmidhuber (1991b); Pathak et al. (2017); Schmidhuber (1991a). Curiosity is built upon the intuition that in unexplored regions of the environment, the forward prediction error of the agent's internal model will be large Schmidhuber (1991b); Pathak et al. (2017). As a result, agents are rewarded for visiting regions of the state space that they have not previously occupied. If, however, a particular state transition is impossible to predict, it will trap a curious agent Burda et al. (2018b); Schmidhuber (1991a). This is referred to as the noisy TV problem (e.g. Burda et al. (2018b); Schmidhuber (1991a)), the etymology being that a naively curious agent could dwell on the unpredictability of a noisy TV screen.

Several existing curiosity-like methods Burda et al. (2018b); Pathak et al. (2017; 2019) aim to avoid noisy TVs or "stochastic traps" Shyam et al. (2019). Nevertheless, employing dynamics based prediction errors as intrinsic rewards is difficult as current methods either fail when stochastic traps are action-dependent, or require an ensemble of dynamics models Pathak et al. (2017; 2019); Shyam et al. (2019); Burda et al. (2018a). This work presents aleatoric mapping agents (AMAs), which uses aleatoric uncertainty estimation Kendall & Gal (2017) to escape stochastic traps.

AMAs are both inspired by and build upon proposals developed within neuroscience Yu & Dayan (2005), that suggest expected uncertainties in top down predictions are signalled by the modulation of cortical acetylcholine. We build on this theory by algorithmically demonstrating a functional purpose for expected *aleatoric* uncertainties that is consistent with, but also further specifies, Yu & Dayan (2005)'s predictions of cholinergic activity in the Posner task. Our work adds further credence to Yu & Dayan (2005)'s call for experimental tests of whether acetylcholine signals epistemic or aleatoric uncertainties. Accordingly, a blueprint for a VR mouse task, along with theoretical predictions of cholinergic uncertainty signalling in such a task, is presented in Section 5.

## 2 BACKGROUND

### 2.1 EPISTEMIC AND ALEATORIC UNCERTAINTIES

The uncertainty of a statistical model can be described as the sum of two *theoretically* distinct types of uncertainty: epistemic uncertainty and aleatoric uncertainty (e.g. Hora (1996), see Hüllermeier & Waegeman (2021) for a review). Epistemic uncertainty measures the errors of a model's prediction that can be minimised with additional experience and learning Hüllermeier & Waegeman (2021). As a result, an agent using epistemic uncertainties as intrinsic rewards tends to value dynamics it has not previously encountered, and hence cannot predict accurately, but could learn to predict in the future (e.g. Osband et al. (2016)). More concretely, epistemic uncertainty can be considered to be the expected information gain of observing the next predicted state Mukhoti et al. (2021). On the other hand, prediction errors that are due to aleatoric uncertainties are, by definition, a result of unpredictable processes Hüllermeier & Waegeman (2021). Therefore, any agents that receives intrinsic rewards for aleatoric dynamics risks being trapped, as exemplified by the noisy TV problem Schmidhuber (1991a); Burda et al. (2018a).

Direct estimation of epistemic uncertainty surrounding future states would be an ideal basis for a curious agent but tractable epistemic uncertainty estimation with high dimensional data is an unsolved problem Gal (2016). We implicitly incentivise agents to seek epistemic uncertainties by removing the aleatoric component from the total prediction error. This is similar to methods that separate epistemic and aleatoric uncertainties in return predictions Clements et al. (2019), or within a latent variable model Depeweg et al. (2018)—allowing for the construction of policies that are rewarded for exploring their environments and punished for experiencing aleatoric uncertainty. However, as far as we are aware, we are the first to compute aleatoric uncertainties within a *scalable* curiosity framework to reduce *intrinsic rewards* for those state transitions with aleatoric uncertainty.

### 2.2 CURIOSITY AND INTRINSIC MOTIVATION IN REINFORCEMENT LEARNING

Curiosity-driven Pathak et al. (2017) agents assign value to states of the environment that they deem to be "interesting" Still & Precup (2012); Schmidhuber (1997). How a curiosity based method computes whether a state is "interesting" Still & Precup (2012); Schmidhuber (1997) is usually its defining characteristic. The original formulation of curiosity used prediction errors directly as intrinsic rewards Schmidhuber (1991b). The noisy TV problem quickly emerged when using this naïve approach in stochastic environments Schmidhuber (1991a). In order to evade the allure of stochastic traps, the first proposed solution to the noisy TV problem implements "interesting" Still & Precup (2012); Schmidhuber (1997) as prediction errors that reduce over time Schmidhuber (1991a); Kaplan & Oudeyer (2007). Others consider "interesting" Still & Precup (2012); Schmidhuber (1997) to mean a high dependency between present and future states and actions (i.e. "interesting" things are predictable Still & Precup (2012) or controllable Mohamed & Jimenez Rezende (2015)).

Inverse dynamics feature (IDF) curiosity Pathak et al. (2017) rejuvenated interest in using one step prediction errors as intrinsic rewards. IDF curiosity avoids stochastic traps by computing prediction errors with features that aim to only contain information concerning stimuli the agent can affect Pathak et al. (2017). Further experiments Burda et al. (2018a) showed that simple one-step prediction errors also work effectively within a random representation space generated by feeding state observations through a randomly initialised network. Burda et al. (2018a) also showed the (IDF) approach is vulnerable to action-dependent noisy TVs—demonstrated by giving the agent a 'remote control' to a noisy TV in the environment that could induce unpredictable environment transitions. This motivated random network distillation (RND) Burda et al. (2018b), which removes dynamics from the prediction problem altogether—instructing a network to learn to predict the output of another fixed randomly initialised network at each state, using the resulting error as intrinsic rewards.

Other exploration methods explicitly leverage uncertainty quantification for exploration. The canonical approach is "optimism under uncertainty", which in its most basic form means weighting the value of state-actions pairs inversely to the number of times they have been experienced (Sutton & Barto, 2018, p. 36). Known as count based methods Strehl & Littman (2008); Bellemare et al. (2016), this approach was shown to reliably evade noisy TVs in minigrid environments Raileanu & Rocktäschel (2020). However, it is not feasible to count state visitations in many environments where there is a large number of unique states Bellemare et al. (2016). "Pseudo-count" methods exchange

tabular look up tables for density models to estimate an analogous intrinsic reward to counts in large state spaces Bellemare et al. (2016). Nevertheless, prediction error based methods for exploration remain a key component in state of the art algorithms deployed in high dimensional state spaces (e.g. Badia et al. (2019)).

Attempts have been made to reward Epistemic uncertainty directly. This typically requires a posterior distribution over model parameters, which is intractable without approximations such as ensembles or variational inference (e.g. Houthooft et al. (2016)). Osband et al. (2016) instantiated an ensemble Lakshminarayanan et al. (2017) approach into the final layer of a deep Q-network—rewarding its agents for epistemic value uncertainty. Pathak et al. (2019) use the variance of ensemble predictions being used as intrinsic rewards, while Shyam et al. (2019) reward experience of epistemic uncertainty within an ensemble of environment models. Lastly, some uncertainty estimation methods have recently been developed that enforce a smoothness constraint in the representation space Mukhoti et al. (2021); van Amersfoort et al. (2021)—allowing for sensible estimations of uncertainty to be made from learned representations—but these approaches have not yet been adopted in reinforcement learning. All in all, uncertainty often plays a role in different formulations of curiosity, which is why we looked at models acetylcholine—a neuromodulator associated with uncertainty—for inspiration in building curiosity models capable of avoiding distractions.

## 2.3 ACETYLCHOLINE

In the mammalian brain acetylcholine is implicated in a range of processes including learning and memory, fear, novelty detection, and attention Ranganath & Rainer (2003); Pepeu & Giovannini (2004); Acquas et al. (1996); Barry et al. (2012); Yu & Dayan (2005); Hasselmo (2006); Giovannini et al. (2001); Parikh et al. (2007). Traditional views—supported by the rapid increase in cholinergic tone in response to environmental novelty and demonstrable effects on neural plasticity— emphasised its role as a learning signal, generating physiological changes that favour encoding of new information over retrieval Hasselmo (2006).

Notably, Yu & Dayan (2003) proposed an alternative perspective, suggesting that acetylcholine signals the *expected uncertainty* of top down *predictions*, while modulation of norepinephrine is a result of *unexpected uncertainties*. More concretelty, Yu & Dayan (2003)'s model can be seen as favouring bottom up sensory input over top down predictions if predictions are believed to be inaccurate—consistent with evidence that shows acetylcholine inhibits feedback connections and strengthens sensory inputs Hasselmo (2006). However this approach does not explicitly separate epistemic and aleatoric uncertainties Yu & Dayan (2005). In contrast, the utility of quantifying epistemic uncertainties for exploration has been widely recognised in the RL literature (e.g. Osband et al. (2016); Pathak et al. (2019)). Here we demonstrate a potential use of aleatoric uncertainties in exploring agents both biological and artificial. Namely, aleatoric uncertainties can be used to divert attention away from unpredictable dynamics when using prediction errors as intrinsic rewards. This is similar to a model proposed by Parr & Friston (2017), suggesting acetylcholine may indicate expected uncertainties in top down predictions within an MDP.

In this context we propose an extension to Yu & Dayan (2005)'s dichotomy. Specifically, we suggest that in the mammalian brain, cortical acetylcholine signals expected aleatoric uncertainties, while norepinephrine is modulated by epistemic uncertainties both expected and unexpected. This formulation is attractive in an ML framework, providing a means to avoid stochastic traps, while also being consistent with empirical biological data Hasselmo (2006); Yu & Dayan (2003). Testing this hypothesis calls for experiments to clarify the nature of the uncertainty signalled by acetylcholine, which we present in Section 5.

## 3 METHOD

AMAs operate in the arena of Markov decision processes that consist of states $s \in \mathcal{S}$, actions $a \in \mathcal{A}$, and rewards $r \in \mathcal{R} \subset \mathbb{R}$ Sutton & Barto (2018). At each timestep $t$ the agent selects an action via a stochastic policy $a_t \sim \pi(\cdot|s_t)$ Szepesvári (2010) and then receives a reward $r_{t+1}$ and state $s_{t+1}$ generated via the transition function $p(\mathbf{s}_{t+1}, r_{t+1}|\mathbf{s}_t, a_t)$ of the environment Sutton & Barto (2018). The objective of the agent is to learn a stochastic policy $\pi$, parametrised by $\xi$, that generates an action as a function of the current state $s_t$, which aims to maximise the expectation of the sum of discounted

future rewards (e.g. Mnih et al. (2016)).

$$\max_{\pi_\xi} \mathbb{E}_{\pi_\xi} \left[ \sum_{k=0}^{T} \gamma^k r_{t+k} \right] \tag{1}$$

Where $T$ is the episode length and $\gamma$ is the discount factor. Following other curiosity based methods, the total reward is the sum of the intrinsic reward provided by the intrinsic reward module of the agent and extrinsic rewards provided by the environment (e.g. Pathak et al. (2017); Badia et al. (2019); Raileanu & Rocktäschel (2020); Burda et al. (2018b)).

$$r_t = \beta r_t^i + r_t^e \tag{2}$$

Where the superscripts $i$ and $e$ indicate intrinsic and extrinsic rewards, and $\beta$ is a hyperparameter that regulates the influence of intrinsic rewards on the policy. In previous works Burda et al. (2018a), the intrinsic reward $r_t^i$ is equal to the mean squared forward prediction error of a curiosity module. To avoid stochastic traps we subtract the aleatoric uncertainty—which is constrained to have a diagonal covariance Kendall & Gal (2017)—from the prediction error, so that agents are not surprised by transitions that were previously learnt to be unpredictable.

$$r_t^i = \|\mathbf{s}_{t+1} - \hat{\mu}_{t+1}\|^2 - \eta \operatorname{Tr}(\hat{\boldsymbol{\Sigma}}_{t+1}) \tag{3}$$

Where $\hat{\mu}_{t+1}$ is the predicted mean of the next state, $\hat{\boldsymbol{\Sigma}}_{t+1}$ is the predicted aleatoric uncertainty of the next state and $\eta$ is a hyperparameter that regulates by how much the predicted uncertainty of the next state effects intrinsic rewards. The states being predicted are usually image observations but AMAs can also work with learned representations of state observations as we show in Section 4.4. To learn to predict the mean of the next state $\hat{\mu}_{t+1}$ and its aleatoric uncertainty $\hat{\boldsymbol{\Sigma}}_{t+1}$, we follow Kendall & Gal (2017)—fitting a diagonal covariance Gaussian distribution to the elements of the next state. The predictions are made by a double-headed neural network—with a mean prediction head $\mathbf{f}$ parameterised by $\theta$ and a variance prediction head $\mathbf{g}$ parametrised by $\phi$. As employed in previous works, the separate heads of the double-headed deep network share feature extracting parameters Kendall & Gal (2017). The prediction network performs *heteroscedastic* aleatoric uncertainty estimation Kendall & Gal (2017), which in a reinforcement learning context means the prediction heads are conditioned on the current state and action.

$$p(\mathbf{s}_{1:N}|\theta, \phi) = \prod_{t=1}^{N} \mathcal{N}(\mathbf{s}_{t+1}; \mathbf{f}_\theta(\mathbf{s}_t, \mathbf{a}_t), \mathbf{g}_\phi(\mathbf{s}_t, \mathbf{a}_t)) \tag{4}$$

Where $N$ is the total number of states observed during training. While Kendall & Gal (2017) use *maximum a posteriori* inference with a zero-mean Gaussian prior on the network parameters $\{\theta, \phi\}$, we found empirically that the resulting regularisation terms in the cost function did not improve results. Accordingly, we simply perform maximum likelihood estimation with the likelihood presented in Equation (4), resulting in the following cost function Kendall & Gal (2017).

$$\mathcal{L}_{t+1}(\theta, \phi) = (\mathbf{s}_{t+1} - \hat{\mu}_{t+1})^\top \hat{\boldsymbol{\Sigma}}_{t+1}^{-1} (\mathbf{s}_{t+1} - \hat{\mu}_{t+1}) + \lambda \log(\det(\hat{\boldsymbol{\Sigma}}_{t+1})) \tag{5}$$

The first term is the familiar mean squared error *divided by the uncertainty*. The second term blocks the explosion of predicted aleatoric uncertainties Kendall & Gal (2017). We follow Kendall & Gal (2017)'s prescription of estimating $\log \boldsymbol{\Sigma}$ instead of $\boldsymbol{\Sigma}$ to ensure stable optimisation. Furthermore, the hyperparameter $\lambda$ was added tempering the model's aleatoric uncertainty budget (e.g. Depeweg et al. (2018); Clements et al. (2019); Eriksson & Dimitrakakis (2019)). We use the predicted mean and aleatoric uncertianty of the next state—which are being learned online with Equation (5)—to compute intrinsic rewards according to Equation (3). Lastly, we would like to highlight that the policy network is separate to the state prediction network as in other curiosity based methods Pathak et al. (2017).

## 4 EXPERIMENTS

This section a range experiments with noisy TVs. Extra details such as the hyperparameters and architectures used are in the appendix. Shaded regions are standard error of the mean.

## 4.1 NOISY MNIST

First we completed a supervised learning task, similar to the noisy MNIST environment introduced by Pathak et al. (2019). The environment does not elicit any actions from an agent. Instead, the prediction network simply needs to learn one step mappings between pairs of MNIST handwritten digits. The first images in the pairs are randomly selected 0s or 1s. When the first image is a 0 then the second image is the exact same image (these are the deterministic transitions). When the first image is a 1, then the second image is a random digit from 2-9 (these are the stochastic transitions). A prediction model capable of avoiding noisy TVs should eventually learn to compute equal intrinsic rewards for both types of transitions Pathak et al. (2019). We trained two different neural networks

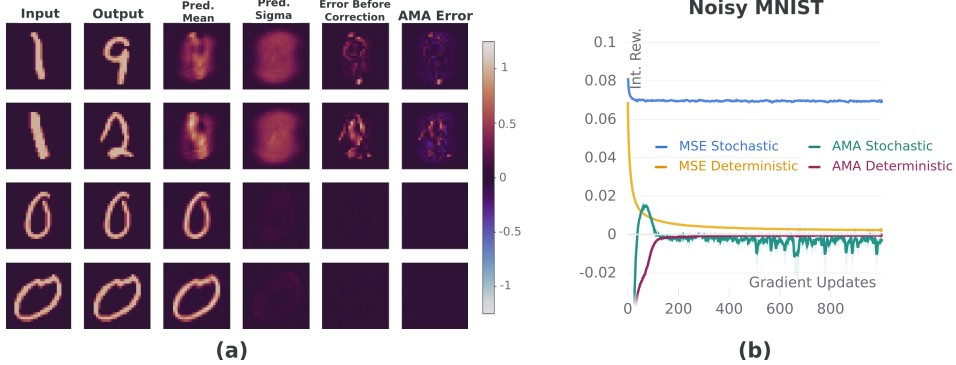

Figure 1: AMAs can learn to ignore stochastic transitions. (a) Example transitions from the Noisy MNIST environment along with associated predictions. The top two rows show stochastic transitions where AMA's predicted variance is high in the majority of the image allowing intrinsic reward to be small despite the stochastic transition. (b) Two reward curves for MSE and AMA are plotted where stochastic is the $1 \rightarrow \{2, ..., 9\}$ transitions and deterministic is the $0 \rightarrow 0$ transitions.

on this task (dapted from Liao (2020)), one with a mean squared error (MSE) loss function—as a baseline— and the other with the AMA loss function Kendall & Gal (2017). The networks are equivalent except that the AMA network has two prediction heads. Both networks contain skip connection from the input layer to the output layer and were optimised with Adam at a learning rate of 0.001 and a batch size of 32 Kingma & Ba (2015). The uncertainty budget hyperparameter $\lambda$ and the uncertainty weighting hyperparameter $\eta$ were set to 1 for the AMA network.

The MSE prediction network is unable to reduce prediction errors for the stochastic transitions, causing it to produce much larger intrinsic rewards than the deterministic transitions, consistent with Pathak et al. (2019). On the other hand, the AMA prediction network is able to cut its losses by attributing high variance to the stochastic transitions, making them just as rewarding as the deterministic transitions.

## 4.2 DEEP LEARNING MINIGRID

Next we test AMA on the Gym MiniGrid environment Chevalier-Boisvert et al. (2018), which allows for resource-limited deep reinforcement learning. The agent receives tensor observations describing its receptive field at each timestep. The channels of the observations represent semantic features (e.g. blue door, grey wall, empty, etc.) of each grid tile. The action space is discrete (containing actions: turn left, turn right, move forward, pick up, drop, toggle objects and done) allowing the agent to move around the environment as wells well as open and close doors. We used singleton environments but with different seeds for each run—resulting in different room configurations—to generate the standard error regions for Figure 2. We measure exploration by counting the number of unique states visited throughout training. An action-dependent noisy TV was added, inspired by other minigrid experiments with noisy TVs Raileanu & Rocktäschel (2020), by setting approximately half of the state observation to uniformly sampled integers within the range of possible minigrid values. When the agent selects the 'done' action the noisy TV is activated in the next observation. This is the only effect of the 'done' action.

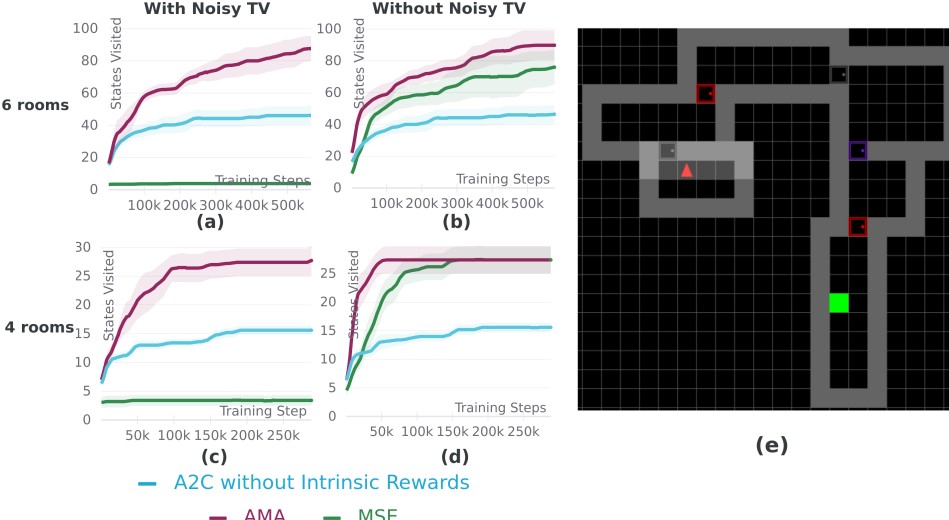

Figure 2: AMA agents effectively explore sparse reward minigrid environments that contain action dependent stochastic traps. (c) and (d) panel show performance on the easiest environment, containing four rooms, while the (a) and (b) show performance on a more challenging environment with six rooms. AMA and MSE have similar exploration performance when the noisy TV is absent, outperforming a no-intrinsic-reward baseline—but when a noisy TV is present only the AMA curiosity approach is able to significantly explore the environment. (e) shows an example six room environment.

We perform policy optimisation with a synchronous advantage actor critic (A2C) implementation recommended by the gym minigrid README Mnih et al. (2016); Willems (2020). For the minigrid experiments we train on intrinsic and extrinsic rewards with their relative weighting being equal. The actor critic weights were optimised with RMSProp Tieleman & Hinton (2014) at a learning rate of 0.001, while the intrinsic reward module was optimised with the Adam optimizer at a learning rate of 0.001 for the AMA agent and 0.0001 for the MSE agent Kingma & Ba (2015). Hyperparameters were optimised for overall performance with and without the noisy TV, see appendix for details. The forward prediction module works in the observation space as opposed to a learned feature space as is implemented in other curiosity driven methods Pathak et al. (2017); Burda et al. (2018b); Raileanu & Rocktäschel (2020). Pixel based curiosity was chosen due to its simplicity. The forward prediction model is a double headed CNN, which builds upon a previous intrinsic motivation implementation on minigrid Raileanu & Rocktäschel (2020). The uncertainty budget $\lambda$ of the AMA network was set to 0.1, the uncertainty weighting $\eta$ was set to 1 and intrinsic rewards were clipped below zero.

We perform experiments in four and six room configurations of the environment. Without a noisy TV both AMA and MSE reward functions generate visit more states compared to the no intrinsic reward baseline. On the other hand, the presence of a noisy TV profoundly affects the performance of the MSE curiosity agent, greatly reducing the number of states visited. In contrast, AMA agents are almost unaffected by the presence of an action dependent noisy TV.

### 4.3 MARIO AND SPACE INVADERS

Next we examine whether our findings from the MNIST and minigrid experiments scale to games from Atari Bellemare et al. (2013) and Gym Retro Nichol et al. (2018); Kauten (2018). We adapt the proximal policy optimisation (PPO) Schulman et al. (2017) curiosity implementation from Burda et al. (2018a), using pixel based in our AMA system. We extend Burda et al. (2018a)'s U-Net Ronneberger et al. (2015) to use two output heads to predict the mean and variance of future states. For our baselines we use official implementations from Burda et al. (2018b) and Burda et al. (2018a). We leave all PPO hyperparameters equal to their values from Burda et al. (2018a). We set the uncertainty budget hyperparameter $\lambda$ to 1 and the uncertainty weighting hyperparameter $\eta$ to 2. For the these experiments we do not clip intrinsic rewards. To simulate the noisy TV thought experiment, we extend the environment's action space with an action that induces grayscale tiled CIFAR-10 Krizhevsky et al.

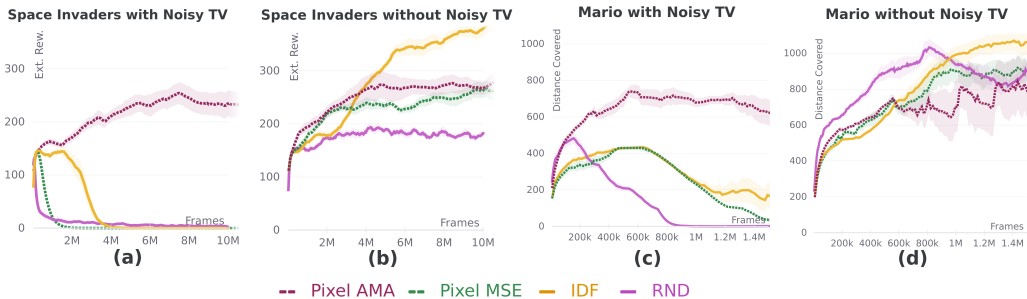

Figure 3: Pixel AMA performs significantly better than all baselines with a noisy TV (a) and (c) and without the distracting noisy TV AMA nearly matches its most directly comparable method Pixel MSE (b) and (d). No extrinsic rewards were used for policy optimisation. In Mario distance covered and extrinsic reward are equivalent. The y-axis plots extrinsic rewards per episode.

(2014) images (examples in appendix) in place of game frames for the next observation and sends the ultimate action of the action space to the game emulator. The choice of the ultimate action is arbitrary. In the appendix we also demonstrate robustness (with the same hyperparameters) to uniform noise.

We compare AMA to three alternative intrinsic reward methods: random network distillation (RND) Burda et al. (2018b), inverse dynamics feature (IDF) curiosity Pathak et al. (2017) and MSE pixel based curiosity Burda et al. (2018a). *Policy optimisation is done with intrinsic rewards only but we measure performance with extrinsic rewards* following Burda et al. (2018a). Compared to the relatively weak baseline of pixel based curiosity, the Space Invaders and Mario experiments show similar results to the minigrid experiments—MSE and AMA pixel based curiosity have comparable performance when no noisy TV is present (Figure 4(b) and 4 (d)), while with the noisy TV AMA greatly outperforms MSE pixel based curiosity (Figure 4(a) and Figure 4(c)). Unsurprisingly, random network distillation and inverse dynamics curiosity maintain their superiority over pixel based methods without a noisy TV (Figure 4(b) and 4(d)). However, unlike AMA curiosity, both these baselines are vulnerable to action dependent noisy TVs (Figure 4(a) and Figure 4(c)). While RND has previously been shown to be able to evade stochastic traps Burda et al. (2018b), it seems that if the number of novel states in the trap is large then RND remains vulnerable to trapping.

## 4.4 BANK HEIST: AN ATARI GAME WITH A NATURAL STOCHASTIC TRAP

Although Atari games are famously deterministic Machado et al. (2018), we identified a naturally ocurring stochastic trap in the Bank Heist gameplay videos of the original IDF curiosity paper. The objective of Bank Heist is to simultaneously avoid police cars and navigate to banks distributed across four 2D mazes. The mazes can be entered and exited through the sides of the screen. With each enter/exit the bank locations reset stochastically. Additionally, the getaway car has the option to drop dynamite in order to destroy police cars.

When trained on purely intrinsic rewards, IDF curiosity will perpetually enter and exit the maze while also dropping dynamite. This creates high prediction error as it is impossible to predict when the dynamite will explode and where the banks will regenerate. Coincidentally, an agent stuck in this stochastic trap achieves a high extrinsic reward—resetting the bank positions causes banks to occasionally regenerate on top of the getaway car increasing game score but is it actually exploring? We measured the average number of maze pixels crossed by the agent's car in an episode. In doing so, we see that as extrinsic rewards rise—i.e the point at which the agent begins to regenerate the banks—the average coverage of the maze sharply decreases.

Next we integrated the AMA prediction paradigm into the IDF approach, predicting the mean and aleatoric variance of future state representations and computing rewards with Equation (3) with AMA hyperparameters $\lambda$ and $\eta$ set to 1. With the AMA approach the car visited more positions in the maze as it not susceptible to the stochastic trap but performed poorly in terms of extrinsic rewards as more exploration does not necessarily correlate with better extrinsic reward.

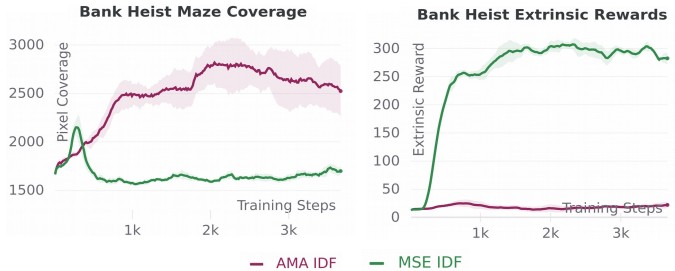

Figure 4: Pixel AMA performs significantly better than all baselines with a noisy TV (a) and (c) and without the distracting noisy TV AMA nearly matches its most directly comparable method Pixel MSE (b) and (d). No extrinsic rewards were used for policy optimisation. In Mario distance covered and extrinsic reward are equivalent. The y-axis plots extrinsic rewards per episode.

This section demonstrates that noisy TVs can be an inherent issue even in environments with simple dynamics. Further, the results highlight the pitfalls of measuring exploration performance with extrinsic reward (which the agent is not trained to optimise)—although we believe extrinsic reward correlates well with coverage for Space Invaders and Mario.

## 5 A Proposed Test for the Aleatoric Model of Acetylcholine

Inspired by Yu & Dayan (2005), we propose that in the mammalian brain acetylcholine signals aleatoric uncertainty surrounding future states. However, we are not aware of any experimental neuroscience data that elucidates the specific nature of the uncertainty signalled by acetylcholine. As a result, this section proposes a 1D rodent VR task designed to test the specific nature of cholinergic uncertainty signalling in the mammalian brain which we hope will be picked up by experimental neuroscientists. To supplement our experimental proposal, we compute theoretical predictions of cholinergic activity within either an aleatoric or epistemic acetylcholine model—two competing interpretations of Yu & Dayan (2005)'s work. The aleatoric model uses aleatoric uncertainties as a theoretical acetylcholine signal Kendall & Gal (2017), while the epistemic model uses ensemble variance as an acetylcholine signal Pathak et al. (2019).

The proposed task places an animal in a VR corridor containing a series of spatial landmarks and two reward zones in which it must respond in order to have a chance of receiving a reward. Responding in reward zone A causes the animal to teleport to a random position along the track. Responding in reward zone B causes the animal to teleport to a fixed position on the track.

To compute how both models predict the cholinergic signal should respond in the proposed rodent VR experiment, we simulate the task with a simple multi-armed bandit environment. In our bandit model of the task an agent predicts a 1D function by sampling minibatches from different regions of the input. In one region of sample space the function takes a simple sinuisoidal form, analogous to zone A of the VR track, in a second region the function consists of points randomly sampled from a standard normal distribution at each timestep, analogous to zone B (Figure 5(c)). As described previously, we applied two models to this task, in the first acetycholine was identified with aleatoric uncertainty, while in the second—as a control—acetylcholine tracks epistemic uncertainty.

We trained an action value based multi-armed bandit to maximise intrinsic rewards for two kinds of forward prediction models: a double headed network trained to optimise the AMA objective and an ensemble of networks where each member is minimising their own MSE (e.g. Pathak et al. (2019)). The aleatoric model uses the AMA reward function whereas the epistemic model is (intrinsically) rewarded for variance in ensemble predictions. We plot both models' uncertainties in each reward zone over time—recovering a clear prediction of cholinergic activity in both cases. The aleatoric uncertainty of AMA remains high in reward zone B but decreases in reward zone A. On the other hand, the epistemic model shows a decrease in acetylcholine in both reward zones over time. We hope these clear and distinct predictions on the nature of cholinergic uncertainty signalling will be tested by the experimental neuroscience community in a task similar to the one we propose.

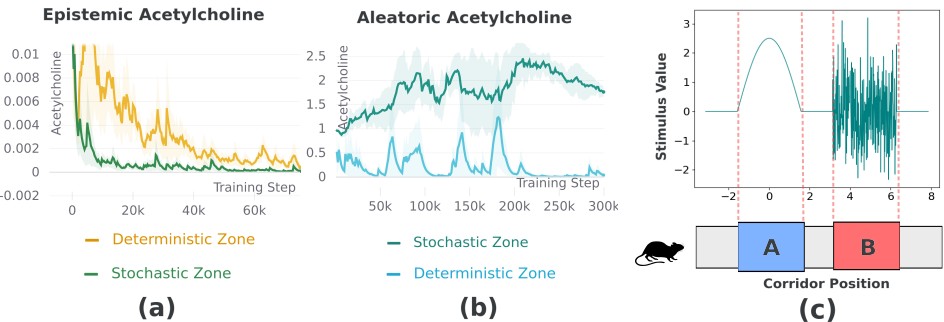

Figure 5: Predictions on a theoretical experiment to illuminate the epistemic or aleatoric nature of cholinergic signalling in the brain. (a) and (b) show that the epistemic model predicts acetycholine will eventually decrease in both zones while the aleatoric model predicts acetylcholine decreasing to zero in the stable zone but remaining high in the noisy zone. Panel (c) shows our 1D model of the proposed animal experiment where the bandit samples different 'corridor positions' and receives scalar stimuli, which it is trying to predict—using the resulting prediction errors as intrinsic rewards.

## 6 LIMITATIONS

The AMA reward function implicitly rewards epistemic uncertainty by assuming the total uncertainty can be decomposed into epistemic and aleatoric uncertainties. While theoretically true Kendall & Gal (2017), there is no guarantee that AMAs are able to surgically subtract those errors due to aleatoric dynamics from the total prediction error. Additionally, aleatoric uncertainty estimates are not guaranteed to be reliable for out of distribution data, meaning intrinsic rewards could become less reliable the further the agent travels into novel territory Mukhoti et al. (2021). In practice we find that a stochastic policy—and clipping in the case of minigrid—offsets potentially deceptive intrinsic rewards. Furthermore, trust region methods in the retro games (i.e. using PPO Schulman et al. (2017) instead of A2C Mnih et al. (2016)) may also compensate for ocassionally deceptive rewards—suggested by the fact that intrinsic reward clipping was not necessary for the retro game experiments. Finally, we note that like all curiosity approaches (e.g. Burda et al. (2018a)), our method generates non-stationary rewards, which is known to make learning difficult for RL agents.

## 7 CONCLUSION

We have shown AMAs are able to avoid action-dependent stochastic traps that destroy the exploration capabilities of conventional curiosity driven agents Burda et al. (2018a). AMAs tractably avoid stochastic traps by decreasing intrinsic rewards in regions with high estimated aleatory. The success of AMAs further cements a proposal from theoretical neuroscience Yu & Dayan (2005) that experimentalists should complete experiments to clarify the nature of uncertainty signalled by acetylcholine. Accordingly, we present an experimental design along with theoretical predictions of cholinergic activities from competing models. Completing the animal experiment proposed has the potential to benefit neuroscience and reinforcement learning alike Hassabis et al. (2017). Future work could also analyse an alternative hypothesis that novelty based methods such as random network distillation could be an accurate model of acetylcholine Barry et al. (2012); Burda et al. (2018b). Future RL research should aim to integrate the AMA approach into curiosity methods that operate in feature spaces besides pixels or even within those methods that circumvent dynamics altogether (e.g. Burda et al. (2018b)), with the aim of achieving state of the exploration even when noisy TVs are present.

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

# 8 APPENDIX

## 8.1 NOISY MNIST

We use three random seeds for the repeats of the MNIST experiments. The results in the graph show test set performance. The hyperparameters used are listed below. The learning rate was manually tuned so that the identity transformation was learned for the deterministic transitions (hence very low loss for the MSE and AMA network) and the AMA network produced sensible uncertainty estimates for the stochastic transitions.

| Hyperparameter | Value |
|---|---|
| MSE Learning Rate | 0.001 |
| AMA Learning Rate | 0.0001 |
| Batch Size | 32 |
| AMA uncertainty budget $\lambda$ | 1 |
| AMA uncertainty coefficient $\eta$ | 1 |

Table 1: Noisy MNIST hyperparameters

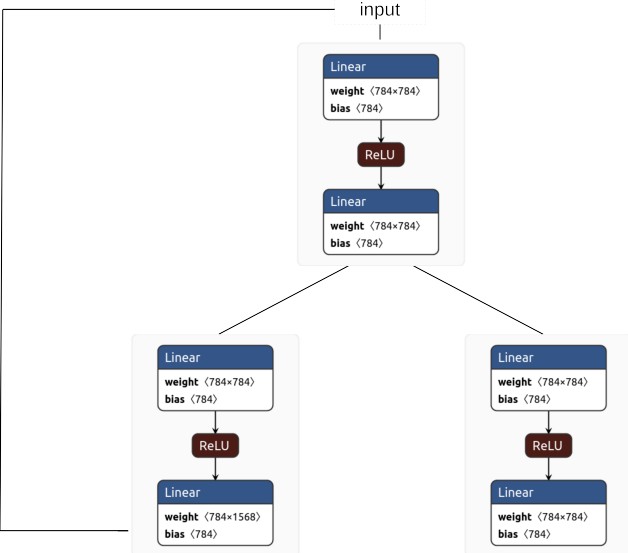

Figure 6: MNIST image to image architecture. For the MSE network the second log variance branch was discarded. A skip connection was provided from the input to the final layer for both AMA and MSE.

## 8.2 MINIGRID

We used 5 seeds for the minigrid experiments. We tuned 2 different hyperparameter via grid search: curiosity learning rate $\in [0.01, 0.001, 0.0001]$ and intrinsic reward scaling (before normalisation) $\in [1, 10, 100]$ for the AMA and MSE modules on the six room environment (by summing up the novel states visited with and without a noisy TV). We used different seeds for the grid search and the final results. We also adapt an implementation of the Welford algorithm from stack overflow for normalising rewards [1]. The architecture for forward prediction is adapted from the implementation from Raileanu & Rocktäschel (2020) but in preliminary experimentation we ended up changing their prediction architecture dramatically.

---

[1]https://stackoverflow.com/a/5544108/13216535

| Hyperparameter | Value |
|---|---|
| AMA learning rate | 0.001 |
| MSE learning rate | 0.0001 |
| AMA reward scaling | 1 |
| MSE reward scaling | 1 |
| AMA normalise rewards | True |
| MSE normalise rewards | False |
| RMS Prop $\alpha$ | 0.99 |
| RMS Prop $\epsilon$ | 1.000e-8 |
| number of actors | 16 |
| unroll length | 5 |
| discount factor $\gamma$ | 0.99 |
| policy learning rate | 0.001 |
| GAE $\lambda$ | 0.95 |
| entropy coefficient | 0.01 |
| value loss coefficient | 0.5 |
| max grad norm | 0.5 |
| AMA uncertainty budget $\lambda$ | 0.1 |
| AMA uncertainty coefficient $\eta$ | 1 |

Table 2: Minigrid hyperparameters

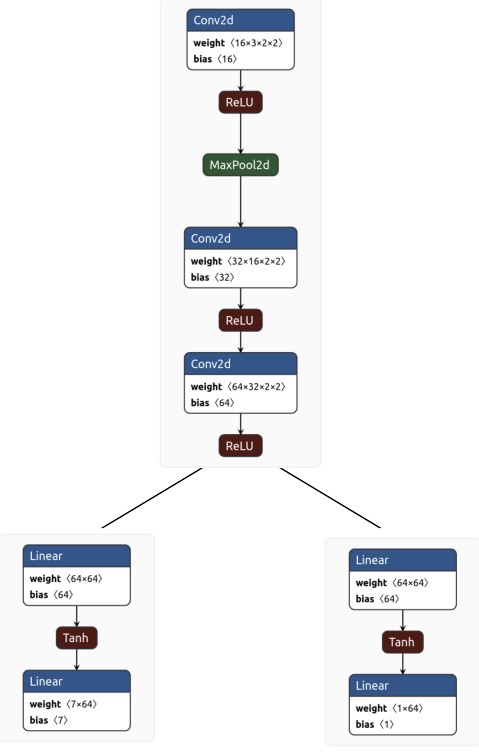

Figure 7: Actor critic architecture for the policy network in the minigrid experiments.

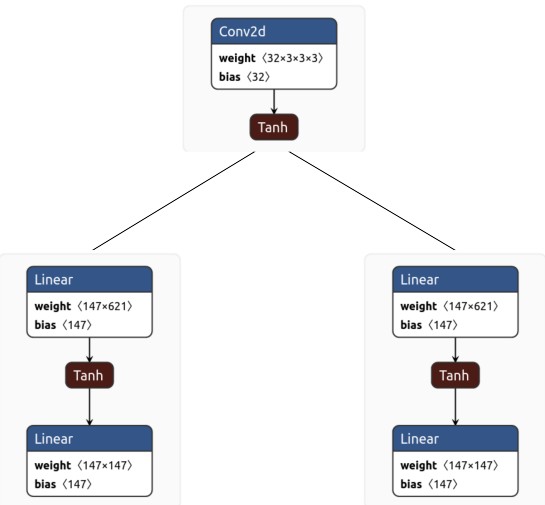

Figure 8: Curiosity forward prediction architecture for the minigrid experiments. For the MSE baseline the variance predictions are not used and loss is computed via a standard MSE.

| Hyperparameter | Value |
|---|---|
| global learning rate | 0.0001 |
| normalise rewards | True |
| number of PPO epochs | 3 |
| number of actors | 128 |
| unroll length | 128 |
| discount factor $\gamma$ | 0.99 |
| GAE $\lambda$ | 0.95 |
| entropy coefficient | 0.001 |
| value loss coefficient | 0.5 |
| policy gradient clip range | [-1.1, 1.1] |
| Pixel AMA uncertainty budget $\lambda$ | 1 |
| Pixel AMA uncertainty coefficient $\eta$ | 2 |
| IDF AMA uncertainty budget $\lambda$ | 1 |
| IDF AMA uncertainty coefficient $\eta$ | 1 |

Table 3: Retro game policy hyperparameters

## 8.3 ATARI

For each run in the Space Invaders and Mario experiments we used five seeds per method. For the Bank Heist experiments we used three seeds per method.

We use the official implementations Burda et al. (2018b;a) for the baselines we compare to (with their default hyperparameters). For AMA and Pixel MSE we adapt from Burda et al. (2018a). The hyperparameters used for our AMA experiments can be found in Table 3. We did not change the PPO/vanilla curiosity hyperparameters from the original implementation we adapted and only changed the AMA hyperparameters. The hyperparameters were chosen by first exploring different configurations on smaller minigrid environments and evaluating promising configurations on the Atari environments.

The UNet architecture used for the forward predictions is described below. We duplicate the decoder head to create a two headed output but we only describe the encoder and decoder here. For the Pixel MSE baselines we use identical architectures but don't use the uncertainty predictions and train on MSE only. There are UNet style residual connections between the corresponding encoder and

| Layer Type | Filters | Kernel Size | Stride |
|---|---|---|---|
| Conv2d | 32 | 8 | 3 |
| Conv2d | 64 | 8 | 2 |
| Conv2d | 64 | 4 | 2 |
| Dense (512 Units) | N/A | N/A | N/A |
| Conv2d Transpose | 64 | 4 | 2 |
| Conv2d Transpose | 32 | 8 | 2 |
| Conv2d Transpose | 4 | 8 | 2 |

Table 4: Retro game forward prediction hyperparameters

| Hyperparameter | Value |
|---|---|
| epistemic learning rate | 0.0001 |
| epistemic batch size | 32 |
| aleatoric learning rate | 0.001 |
| aleatoric batch size | 1000 |
| aleatoric uncertainty budget $\lambda$ | 1 |
| aleatoric uncertainty coefficient $\eta$ | 1 |
| $\epsilon$ greedy $\epsilon$ | 0.1 |

Table 5: Bandit hyperparameters.

decoder layers. Leaky ReLU activations are used in the encoder layers and Tanh activations are used in the decoder layers. Batch normalisation is used throughout the hidden layers. Action information is concatenated at each layer. See supplementary code for further details.

To integrate AMA into the IDF approach, we did not share any representations between mean and variance prediction heads, instead we used two prediction MLPs for the mean and variance. Leaky ReLU is used throughout hidden layers and action information is concatenated at each layer. We used five layers with 512 units each and UNet style residual connections.

## 8.4 BANDIT

We performed 3 repeats to produce the standard error regions show in the graph. Learning rate was tuned by hand, observing how well the network performed in making predictions as the bandit sampled different regions of the environment. The intrinsic reward method for the epistemic bandit is based on Pathak et al. (2019); Lakshminarayanan et al. (2017). We use an action value based bandit algorithm with $\epsilon$-greedy exploration (Sutton & Barto, 2018, p. 31).

## 8.5 PLOT FORMATTING

For the plots we use Weight and Biases[2] including their built in smoothing function. For all results we use the exponential moving average (with a smoothing parameter of 0.59) except in the minigrid plots where we used the moving average smoother (with a smoothing parameter of 0.1).

---

[2](https://wandb.ai/)

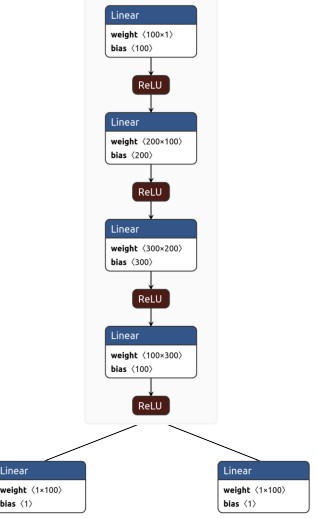

Figure 9: AMA prediction network for bandit tasks.

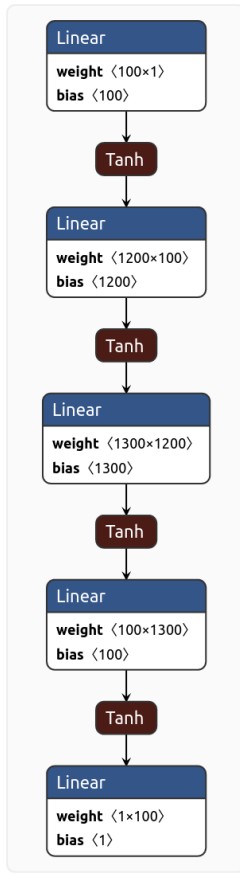

Figure 10: Epistemic prediction network for the bandit task.

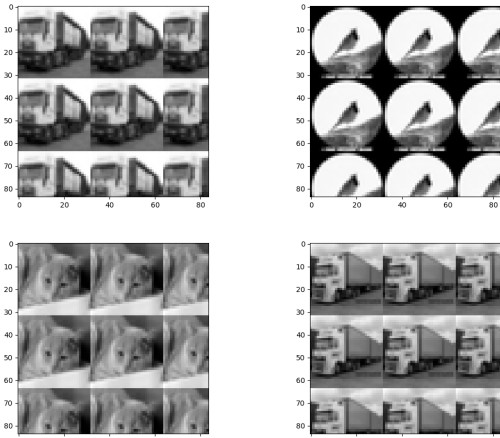

Figure 11: Four examples of frames that an agent might see when interacting with the CIFAR noisy TV. This is a complex noise distribution picked to test the limits of the heteroscedastic aleatoric uncertainty estimation.

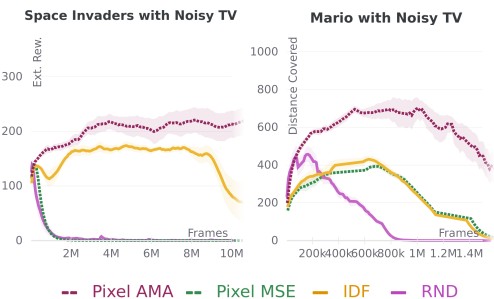

Figure 12: AMA is robust to Noisy TVs of a very random unifrom noise distribution (random pixels from 0-255) while other baselines are also trapped by this additional noisy TV that is significantly different to the noise distribution used for the results used in the main text.

## 8.6 DIFFERENT NOISE DISTRIBUTION FOR RETRO GAMES

For the CIFAR noisy TV we tiled a random CIFAR image (from the training set) for each frame observed on the noisy TV. This required around around 2.5 tiles to fill the $84 \times 84$ pixels of the retro game frames. An example frame can be seen in Figure 11.

## 8.7 HYPERPARAMETER ANALYSIS

The AMA curiosity method contains two hyperparameters $\lambda$ and $\eta$. We analysed on Mario, Space Invaders and Minigrid how sensitive performance is to the values of these hyperaparameters. We find that decent results can be achieved with no tuning on the retro game environment (Figure 14 and Figure 15), while in the six room minigrid environment it is important to tune $\lambda$ (Figure 16).

## 8.8 EVENTUAL DECREASE IN EXPLORATION DURING MARIO TRAINING

In initial experiments we noticed the AMA agent lost motivation to explore its environment after reaching it peak extrinsic reward. To ensure that this was not an inherent problem with AMA, we

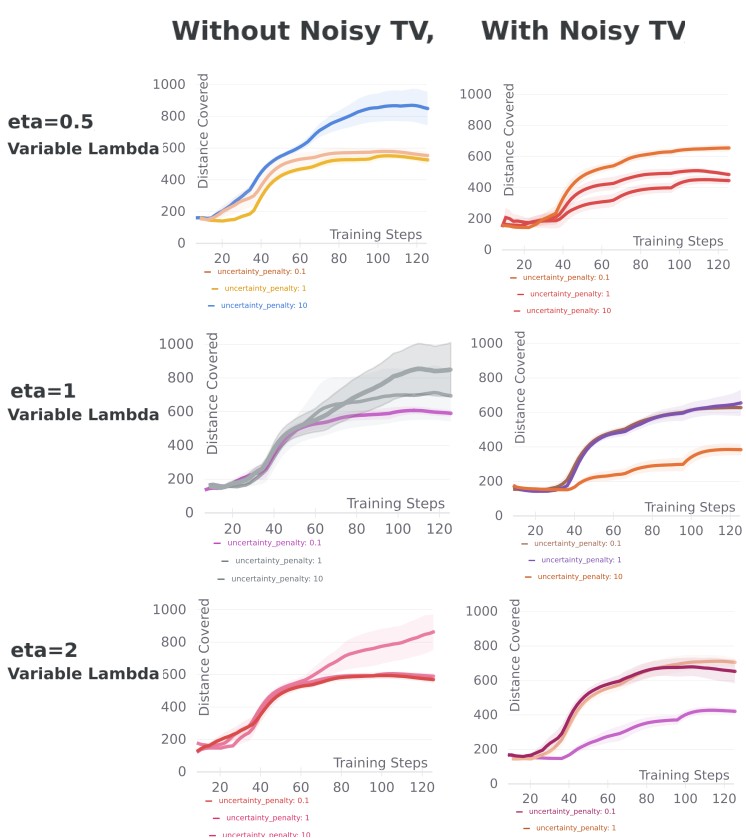

Figure 13: Overall, the mario results are not very sensitive to the setting of $\eta$.

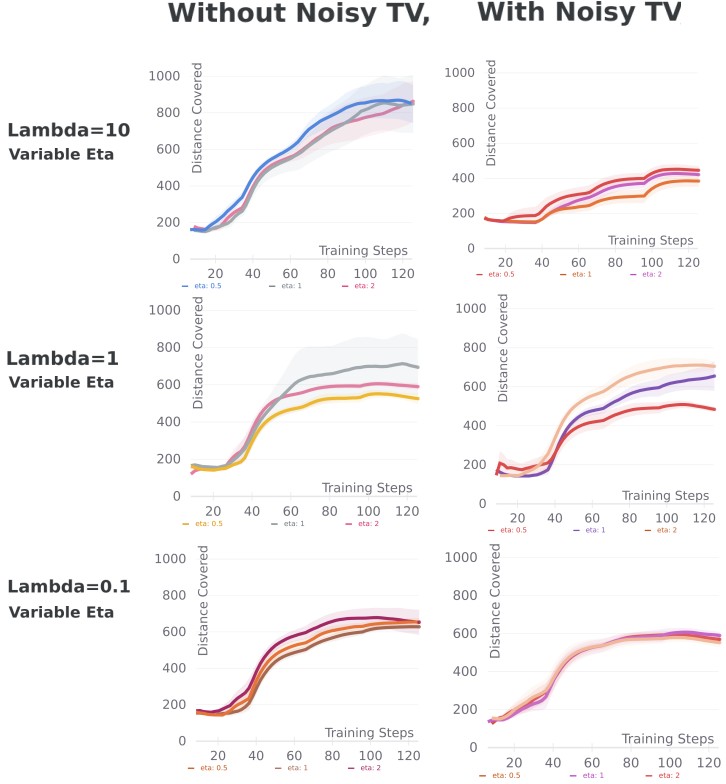

Figure 14: As the uncertainy penalty ($\lambda$) increases performance with the noisy TV becomes poor (expected) because higher uncertainy penalty moves AMA more towards a MSE curiosity. Otherwise for Mario results are not very sensitive to hyperparameter values.

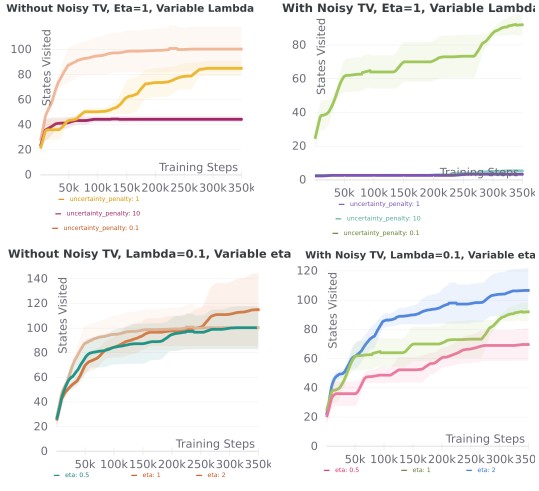

Figure 15: Performance on minigrid is not very sensitive to $\eta$ but $\lambda$ requires some tuning.

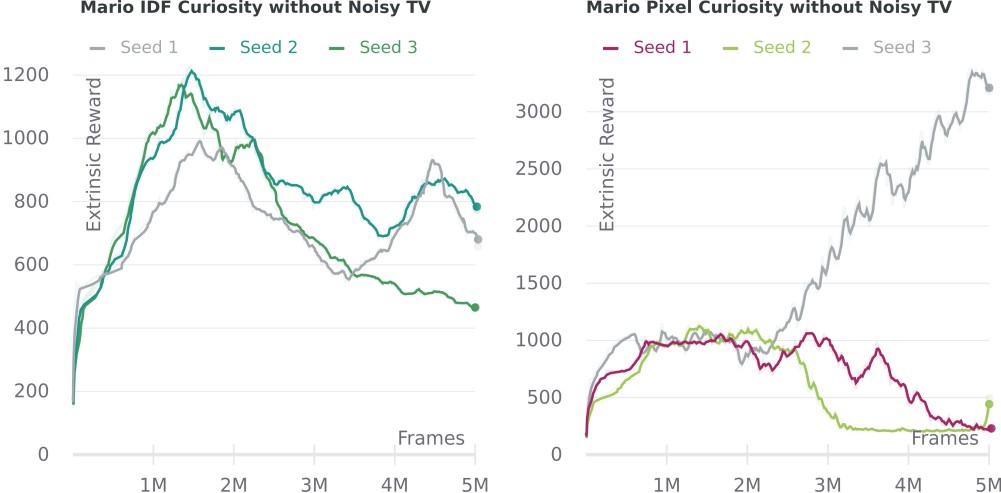

Figure 16: Repeats ran for more frames on Mario for MSE Pixel curiosity and MSE IDF curiosity. 3/3 of seeds see a decrease in performance for IDF curiosity and 2/3 in the pixel curiosity case.

ran the other curiosity methods for further frames and found similar eventual decreases in extrinsic reward (Figure 16). Presumably the cause of this is that once a significant portion of the environment has been explored the agent is no longer motivated to return to those regions (as prediction error decreases)—this issue has been noted by previous authors Pathak et al. (2017).

### 8.9 POTENTIAL NEGATIVE SOCIAL IMPACTS

The work presented is very far from any real world deployment. If it were to be deployed in any real world context then extensive testing would need to be done to understand how the curiosity agents would behave in novel environments as erratic behaviours could be dangerous in, for example, a robotic control context. The AMA objective is overarching (like other curiosity methods) and so care should be taken if deploying in something like a recommender system as the agent could find certain behaviours intrinsically rewarding that you might not have intended it to (like the noisy TV problem). Lastly, although the AMA system contains notions of uncertainty quantification, that does not mean it is able to completely understand the limits of its predictions and so one should not be overconfident in its abilities to do so.

### 8.10 HARDWARE

The experiments were performed on three different machines depending on their availability: A 32 core CPU with one GeForce GTX TITAN X, a 12 core CPU with two GeForce GTX TITAN Xs and one 8 core CPU with two GeForce 2080Ti GPUs.

We list times here on the 2080Ti machine, the other machines were as much as $2\times$ slower. The minigrid experiments took around 40 minutes per run, the Space Invader experiments took around 12 hours per run, the Mario experiments took around 1 hour and 20 minutes per run.

### 8.11 LICENSING

The repository from Willems (2020) has an MIT license. The code used from Chevalier-Boisvert et al. (2018) has an Apache License 2.0. Raileanu & Rocktäschel (2020) has a creative commons license. Besides those listed we are not aware of any further code licensing. We adapted a copyright free rat silhouette image for the bandit figure [3].

---

[3]https://pixabay.com/vectors/rat-rodent-silhouette-gold-chinese-5184465/

## 8.12 FURTHER CODE ACKNOWLEDGEMENTS

Although we did not directly use their code we would like to acknowledge the following open source contributions that provided a useful reference when implementing Kendall and Gal's Kendall & Gal (2017) aleatoric uncertainty estimation algorithms:

```
https://github.com/ShellingFord221/My-implementation-of-What-Uncertainties-
Do-We-Need-in-Bayesian-Deep-Learning-for-Computer-Vision
```

https://github.com/pmorerio/dl-uncertainty

https://github.com/hmi88/what

