# OpenReview forum: "Escaping Stochastic Traps with Aleatoric Mapping Agents"
_ICLR.cc/2022/Conference — ICLR 2022 Submitted_

### Official Review · Reviewer_DE7e · 2021-10-29

**Correctness:** 4
**Technical Novelty And Significance:** 2
**Empirical Novelty And Significance:** 2
**Recommendation:** 5
**Confidence:** 4

**Main Review:**

*Strengths*

The paper provides a very nice, comprehensive review of the existing curiosity techniques, which was a pleasure to read.

I like the neuroscientific inspiration of the proposed algorithm. The field is full of claims about neuro-AI synergy but short of good examples. This work, further developed, may become a good example of such synergy.

The idea of explicitly accounting for aleatoric uncertainty in deciding on future actions seems highly meaningful and, potentially, fruitful.

In this formulation, the uncertainty is estimated with a single agent and not with ensembles of agents, as it was done before. This makes the model more useful.

The authors have performed an impressive set of experiments, involving a significant number of baselines.

I like the idea of proposing a test, which experimental neuroscientists may then use to improve our understanding of the brain.

*Weaknesses*

Provided the close relation of the proposed method to prior work (as mentioned by the authors), it would be reasonable to expect a strong Results part of the paper (see below).

Most of the experiments in this paper were performed in environments specially altered to introduce stochastic traps. The framework’s performance in unaltered environments did not exceed that of the baselines.

The exploration has been previously developed as a tool with a final goal of maximizing rewards in scarce environments. In this work, the authors have reported improvements in exploration but there are no experiments indicating improvement of total reward in standard unaltered environments (the authors do explicitly discuss this issue). It is unclear whether additional exploration, as suggested here, leads to higher rewards in the standard or real-world environment, which is the goal of exploration in the first place.

In the proposed experiment for establishing the role of acetylcholine in the brain, the prediction seemingly boils down to the fact that epistemic uncertainty decays over time, whereas aleatoric uncertainty remains constant. These predictions seem to directly follow from the definitions of such uncertainties, so it is unclear whether the proposed design of the experiment is necessary.

*Specific comments*

In Equation (5), is the last sign (+) correct?

Although the notation is clear, consider introducing the variables before using them.

Whereas the introduction is generally great, it may be useful to talk a bit more about Angela Yu and Peter Dayan’s model, as it forms the main neuroscientific motivation for the proposed framework, so that the readers won’t have to read their paper.

I would personally consider adding a short description of how Equation (5) was derived. It’s pretty clear for me, but may or may not be clear for everyone. The derivation is super simple and won’t take up a lot of space.

*Suggestions to the authors*

I found this work highly promising and would love to see future developments based on your framework. Perhaps, something along these lines:

It would be great to see the improvement of the received reward in standard or real-world environments. The claim here, supported by the literature, is that stochastic traps are the major problem in curiosity-driven exploration. This implies that there are enough standard environments where stochastic traps affect the efficiency of baseline curiosity-based algorithms. Such environments can be used to showcase the benefits of the proposed algorithm when it comes to rewards which, consequently, would skyrocket the impact of this work.

It would also be nice if the author follow up on their proposed acetylcholine experiment and collaborate with experimental neuroscientists to test this hypothesis. Determining that acetylcholine encodes one type of uncertainty or another would be an important result and another great way to ascertain the high impact of this work.


**Summary Of The Paper:**

This paper extends the works on curiosity in artificial agents by incorporating an explicit prediction of (non-reducible) aleatoric uncertainty into the agent’s action choice computation, such that the agents have a preference against the environmental states where uncertainty cannot be reduced by learning. The proposed framework has enabled efficient exploration in a set of tasks where non-reducible uncertainty has been introduced; the authors have also used it to propose a test to clarify the role of acetylcholine – the neurotransmitter related to uncertainty – in the brain.

**Summary Of The Review:**

An impressive amount of work has been done; the writing is mostly easy to follow, and the results seem promising; however, provided that the proposed framework is closely related to the existing ones, further research towards practical results is needed. The better exploration, proposed here, is important, but it may or may not lead to better-performing agents. I, therefore, think that the paper is better suited for a conference workshop rather than for the main track. Thus, overall, I tend to recommend rejection.

---

> ### Author Response · Authors · 2021-11-18
> **Response to DE7e**
>
> We appreciate your constructive criticism and we are glad you found bits of the work "highly promising". Below we respond to your comments. The reviewer comments are in bold while our responses are below the bold text.
>
> **Most of the experiments in this paper were performed in environments specially altered to introduce stochastic traps.**
>
> We performed experiments in environments with specially altered environments because the majority of reinforcement learning test beds are very deterministic and hence do not contain stochastic traps. Nevertheless, in Section 4.4 we present a detailed analysis of the performance of our approach in an environment with a naturally occurring stochastic trap and show that our method performs better than standard IDF curiosity.
>
> Previous high impact works (RND and ICM) only use specially altered noisy TV environments. We do this but we also go the extra mile to demonstrate the performance of our method in an environment with a naturally occurring stochastic traps.
>
> **The framework’s performance in unaltered environments did not exceed that of the baselines.**
>
> Indeed, this is the intended effect of our method. AMAs are a simple add on to intrinsic reward methods that allows for better exploration in stochastic environments. If environments are not stochastic, then there is no reason we should expect it to perform better.
>
> **The exploration has been previously developed as a tool with a final goal of maximizing rewards in scarce environments. In this work, the authors have reported improvements in exploration but there are no experiments indicating improvement of total reward in standard unaltered environments (the authors do explicitly discuss this issue). It is unclear whether additional exploration, as suggested here, leads to higher rewards in the standard or real-world environment, which is the goal of exploration in the first place.**
>
> We agree that exploration eventually leading to more rewards is important. We opted to focus on a specific problem of exploration rather than waste computational resources on hyperparameter tuning the extrinsic reward vs. intrinsic reward coefficient to demonstrate that better exploration leads to better performance. This has already been demonstrated extensively in previous works.
>
> **In the proposed experiment for establishing the role of acetylcholine in the brain, the prediction seemingly boils down to the fact that epistemic uncertainty decays over time, whereas aleatoric uncertainty remains constant. These predictions seem to directly follow from the definitions of such uncertainties, so it is unclear whether the proposed design of the experiment is necessary.**
>
> The point of the experimental design is to aid experimental neuroscientists by providing them with a task that can test our predictions. Your analysis is correct but without our final section experimental neuroscientists do not have a concrete experimental design to test our model.
>
> **In Equation (5), is the last sign (+) correct?**
>
> Yes, we believe so. See Kendall and Gal (2017) equation (5) for confirmation.
>
> **Whereas the introduction is generally great, it may be useful to talk a bit more about Angela Yu and Peter Dayan’s model, as it forms the main neuroscientific motivation for the proposed framework, so that the readers won’t have to read their paper.**
>
> Thank you for this suggestion. Yu and Dayan’s work does provide the main motivation behind our proposed framework. With limited space we did not discuss it as much as we would like to so that we were able to cover all of the relevant ML literature. In the future we will consider adjusting this balance.
>
> **I would personally consider adding a short description of how Equation (5) was derived. It’s pretty clear for me, but may or may not be clear for everyone. The derivation is super simple and won’t take up a lot of space.**
>
> Thank you for the suggestion though we think that it is not necessary to provide this derivation to a ML audience.
>
> **It would also be nice if the author follow up on their proposed acetylcholine experiment and collaborate with experimental neuroscientists to test this hypothesis. Determining that acetylcholine encodes one type of uncertainty or another would be an important result and another great way to ascertain the high impact of this work.**
>
> We agree but we think that an experimental neuroscience section on top of our theoretical work is probably beyond the scope of an ICLR paper.

---

> > ### Comment · Reviewer_DE7e · 2021-11-21
> > **Thank you for your detailed response**
> >
> > I would like to thank the authors for their detailed response.
> >
> > I do agree with some points raised by the authors. With the limited space it may indeed be difficult to provide an extensive literature review or derivations of the equations. Also, the other reviewers have requested a shortening of the derivations, so a longer derivation may not be needed. Maybe a good idea could be to leave the main equations in the main text (as suggested by the reviewer QuEW), move the derivations to the (unlimited-size) appendix, and use the reclaimed space on discussing the existing literature and how the current work is different (as per the reviewer j6wR's suggestion).
> >
> > I also do acknowledge that, as the authors claim, the existing RL testbeds may not be suitable for the issue at hand. That's a good point and it justifies the use of altered environments. Although, as the reviewer j6wR has mentioned, the idiosyncratic design of altered environments in the paper alongside the specific choice of the baselines makes it hard to evaluate the proposed algorithm's impact.
> >
> > It seems that a fair comparison therefore would require adding the algorithms from the papers mentioned by the reviewer j6wR and adding the testbeds from these papers to all algorithms as well. As laborious as it is, these experiments (or a reasonable subset thereof) would be the only meaningful way to establish the proposed algorithm's benefits. Although such additional experiments were suggested by reviewer j6wR, the suggestion seems to have been dismissed by the authors on the grounds of the comparison being unfair due to the compute inefficiency of the baselines. The compute efficiency of the proposed algorithm, if shown, could actually make a strong novelty point for the proposed work, so performing the proposed comparisons may be beneficial for the authors.
> >
> > Finally - a minor point - I disagree with the authors on that better exploration leads to better performance. Whereas it is true in many cases (described in the literature, mentioned by the authors), it's not a universal truth. In section 4.4 of the paper (the Bank Heist task) mentioned by the author in their response, the better exploration in AMA has in fact led to the drastic decrease of the extrinsic reward. Provided that this is the only standard task used by the authors, the message that comes across (as to the usefulness of the proposed approach) is the opposite of what the authors would want it to be. Therefore, it would be beneficial for the paper to show an extrinsic reward increase in at least one of the standard environments - perhaps, from the abundant literature mentioned by the authors in that regard.
> >
> > Considering all the points above I stand by my initial valuation of the paper. I would be happy to reconsider if more comparisons/tests are provided distinguishing current work from prior efforts - along the lines suggested by the reviewer j6wR - but for now, such comparisons have not been done, so it is difficult to evaluate the proposed method.

---

> > > ### Author Response · Authors · 2021-11-24
> > > **Thank you for continued discussion.**
> > >
> > > Thank you for engaging in our work and suggesting ways to improve its current state.
> > >
> > > Could you let us know why you think the environment design is idiosyncratic?
> > >
> > > We take your point that more baselines could improve the robustness of the evaluation of our algorithm. However, we are not sure what you mean by showing that our approach is more computationally efficient than ensembles. We think it is clear that one forward pass of our predictor is more efficient than a forward pass through an ensemble of predictors. Did you mean in terms of sample efficiency?
> > >
> > > We agree that the Bank Heist experiment makes a nuanced point and perhaps that in some environments exploration might not help. However, the success of the naive curiosity in a stochastic trap is a glitch where a stochastic trap just so happens to improve extrinsic rewards and we think it demonstrates how our method can improve agent behaviour. Here is a video we uploaded provided by Burda (2018) showing the pathological behaviour: https://www.youtube.com/watch?v=S4YdZe70XMQ.
> > >
> > > In terms of your last point, we are not sure how to interpret the two statements you make:
> > >
> > > "I also do acknowledge that, as the authors claim, the existing RL testbeds may not be suitable for the issue at hand. That's a good point and it justifies the use of altered environments."
> > >
> > > and
> > >
> > > "Therefore, it would be beneficial for the paper to show an extrinsic reward increase in at least one of the standard environments - perhaps, from the abundant literature mentioned by the authors in that regard."
> > >
> > > If possible, we would really appreciate clarification on this so we can take it into account in further revisions.

---

> > > > ### Comment · Reviewer_DE7e · 2021-11-28
> > > > **Clarifications after re-reading the manuscript, cited literature and the author-reviewer interactions.**
> > > >
> > > > Below, I clarify my points along the lines of the questions raised by the Authors.
> > > >
> > > > **Insufficient overlap with conventional choices of testbeds.**
> > > >
> > > > *Burda et al, '19* in their large-scale study of curiosity-driven learning use UNITY and a large set of ATARI games as their testbeds. *Pathak et al, '19*, also referenced by the authors, similarly use UNITY and a smaller set of ATARI games. In *current work*, the authors use a small subset of ATARI games (non-overlapping with *Pathak et al, '19*); also Noisy MNIST (similar to *Pathak et al, '19*), MiniGrid (similar to *Raileanu and Rocktäschel, '19*) and Bank Heist. Noteworthy, in Noisy MNIST and MiniGrid the authors compare their results to RMS and **not** to the models from the respective papers - so the only comparison which we may use to assess the performance of the proposed method is between AMA, IDF, and Pixel MSE on two ATARI tasks (Space Invaders and Mario).
> > > >
> > > > **Idiosyncratic design of the other (altered) testbeds.**
> > > >
> > > > To show the performance boosts of the proposed algorithm, the authors introduce multiple new testbeds which include noisy TVs. These are different from the ones used in literature, so using them to compare AMA with existing algorithms (not designed to work with noisy TVs) may lead to biased results. Specifically, in ATARI games the authors *extend the environment’s action space with an action that induces grayscale tiled CIFAR-10 images (examples in appendix) in place of game frames for the next observation*. This is in contrast with *Pathak et al, '19* who used Sicky Actions, i.e. *at each step, either the agent’s intended action was executed or the previously executed action was repeated with equal probability*. This is also in contrast with *Burda et al, '19* who only used unaltered ATARI games in the paper. Furthermore, from the authors' description it is unclear what tasks they used in MiniGrid and how these relate to the ones used by *Raileanu and Rocktäschel, '19*. From the figure, it looks like the authors used the *MultiRoomN12S10* task, which is then similar to previous work (needs to be clarified!), however, no comparison to the *Raileanu and Rocktäschel, '19* has been reported.
> > > >
> > > > **Little comparison of the extrinsic rewards**
> > > >
> > > > Most of the results reported by the authors are based on exploration reflected in the *intrinsic* rewards; however, it is the *extrinsic* reward that is the end-goal of the enhanced exploration. Although the authors argue that the *better exploration leads to better performance* and that *this has already been demonstrated extensively in previous works*, their own experiments seem to not support this claim (i.e. the Bank Heist task). Contrary, the other works do use the *extrinsic* reward as a final arbiter of the algorithms' performance (e.g. ATARI experiments by *Burda et al, '19*; UNITY experiments by *Pathak et al, '19*, etc.
> > > >
> > > > **The overall lack of principled comparisons with previous work, in my opinion, makes the paper not ready for publication in the current format.** This can be fixed by using standard testbeds (e.g. ATARI games), standard metrics (e.g. extrinsic rewards), and standard baselines (e.g. these outlined by *Burda et al, '19*) - just like it's done in the rest of the literature in the field. So far the only interpretable experiments (ATARI games with no noisy TVs) show performance below that of the baselines. Therefore, until additional principled experiments are done, I don't see any reason to use the proposed algorithm instead of one of these baselines.

---

> > > > > ### Author Response · Authors · 2021-11-28
> > > > > **Thanks for detailed comments and reading the manuscript again**
> > > > >
> > > > > We really appreciate the time taken to read our paper and think about how it can be improved. We would like to clarify a few points and choices we made based on your comments above.
> > > > >
> > > > > We would like to use the UNITY environment as a baseline but it is not publicly available. We have tried to obtain it from one of the original authors but have had no luck so far. It is probably worth another shot as it is a motivating example for us. In terms of the Atari games we chose, we opted for those that require relatively little compute resources to train.
> > > > >
> > > > > Respectfully, we do disagree that the algorithms we compare to are not developed to work with Noisy TVs. IDF curiosity and RND curiosity were both developed with Noisy TVs in mind. Furthermore, we think that the design of the two noisy TV environments (uniform noise and random CIFAR images) are a simple reproduction of the noisy TV thought experiment.
> > > > >
> > > > > For the minigrid experiments we explain in the text (and figures) that we use the 4 room and 6 room configurations. Specifically, these are registered as "MiniGrid-MultiRoom-N4-S5-v0" and "MiniGrid-MultiRoom-N6-v0". We can try to make this clearer in further revisions.
> > > > >
> > > > > In terms of your final point, we indeed use two of the environments from *Burda et al. 19*. The metric we compare on is extrinsic rewards (as you suggest). Then we use two of the more interesting baselines from *Burda et al. 19* (as you suggest) and include an additional baseline of RND. Perhaps we could have made it clearer, that *distance covered and extrinsic reward are equivalent in Mario*. It seems like one of the main issues is our Noisy TV implementation. We cannot use the UNITY environment. The sticky actions environment is not an action dependent noisy TV nor does it emulate the noisy TV thought experiment that motivates our paper and the ones that came before it. As you explained previously "the existing RL testbeds may not be suitable for the issue at hand. That's a good point and it justifies the use of altered environments." With that in mind, what changes would make our Noisy TV environment suitable for comparisons?

---

> > > > > > ### Comment · Reviewer_DE7e · 2021-11-29
> > > > > > **Re:**
> > > > > >
> > > > > > Thank you for your response.
> > > > > >
> > > > > > Ideally, I would like to see:
> > > > > > 1) The experiments, similar to *Burda et al '19* and similar to your *Space invaders without noisy TVs* and *Mario without noisy TVs* **where your approach (AMA) outperforms the existing approaches** (e.g. Pixel MSE, IDF, and RND) as measured by the extrinsic reward. Ideally, I would suggest accompanying such results with a discussion of which tasks allow for better-than-SOTA performance of your method (AMA), which tasks lead to sub-SOTA performance of AMA, and why.
> > > > > > 2) In the experiments with Noisy MNIST and Minigrid it would be nice to see comparisons with *Pathak et al, '19* and *Raileanu and Rocktäschel, '19* models respectively (also accompanied with a discussion along the lines suggested in Pt.1 above).
> > > > > > 3) After having shown a better-than-SOTA performance, your current experiments with noisy TV will be, in my opinion, well-placed.
> > > > > >
> > > > > > The experiments with Noisy TVs are interesting and important, and there are no issues with them, except they alone deem insufficient to show the merits of the proposed algorithm (AMA). So far, the better-than-SOTA performance of the AMAs has not been shown, so it's unclear why we should use AMA instead of the existing methods pursuing the same goal and performing better on the standard tasks. Therefore I hope that the authors will consider appending their paper with experiments like the ones outlined in this comment to make the paper ready for publication

---

### Official Review · Reviewer_QuEW · 2021-10-30

**Correctness:** 2
**Technical Novelty And Significance:** 4
**Empirical Novelty And Significance:** 4
**Recommendation:** 6
**Confidence:** 5

**Main Review:**

I am surprised that this idea was not explored in an earlier paper. Taking the prediction error minus the variance looks like the very first solution to try in order to prevent agents to stay watching a noisy TV. While my knowledge to this domain is limited, to my mind this is the first paper suggesting this idea. Probably other reviewers will prove me wrong.


I expect this approach to work well, and I trust the good experimental results exposed in the paper. I believe this could have a nice impact on the curiosity/exploration efforts in the RL community.


However, there are many weaknesses that makes this paper not ready for a publication:

First, all the mathematical justifications are unclear and slovenly written with too much errors in equations:

Eq 3) is wrongly reported. It should be $S_{t+1}$ in the left of the noise term, and $\hat{S_{t+1}}$ in the right of the variance term.

In Eq 5), $\mu_{S_{t+1}}$ should actually be $S_{t+1}$ (except it's an average over a batch of states $S_{t+1}$ with the same previous $S_t$, but I guess it's not)

In Eq 6) $\hat{S}$ should be $\hat{\mu}$ for consistency.

Also I feel that there is a lot of paragraphes just to describe a very simple gaussian model used to fit $S_{t+1}$ given $S_{t}$ by maximum log-likelihood (in Eq 5). Sigma is the estimated variance, and mu is the estimated mean. Eq 5 is just the log of Eq 4 (with lambda = -1/pi).
Then in Eq 7, it's just R = error - estimated variance. No need for $\eta$, the point is that these terms should have the same expectation at convergence of the fitted model with $\eta=1$.


The bio-inspired aspect needs more content to be convincing:

for example, reporting models or experimental results from the biological papers (for ex, the actual original curves that fig 5 is trying to reproduce).

What I suggest:

The experimental section looks good to me. But all the method section needs to be re-written from scratch. After introducing the MDP notations, I would directly start with the final equation 7: we take the error minus the variance.

Then, show that the mean of this bonus is zero when the model is learned (so whatever is the unpredictability, the bonus will go to zero).
Optionally, look at the sub-optimality behaviour of the bonus: is the error higher or lower than the variance? How does it affect the agent's curiosity?

Finally explain how the mean and variance are learned, here with a two-headed network trained by maximum LLH.
Also, discuss what happen if the action are taken into account in the learned model (so far the bonus is on-policy and this should appear somewhere and be discussed). Here everything works as long as the agent is not changing his policy, but depending on the learning dynamics of the policy, the model may never converge, this would not happen with action taken into account.





**Summary Of The Paper:**

This paper suggests an intrinsic bonus for exploration that avoids noisy TV by adding a penalty for the estimated variance of the reached state $S_{t+1}$ given previous state $S_{t}$.

For this, they fit an independent normal model of the new state $S_{t+1}$ with mean ($\mu_{t+1}$) and var ($\sigma_{t+1}^2$) predicted from the previous state $S_{t}$.
Then, the bonus is given by the error of prediction $(S_{t+1} - \hat{\mu_{t+1}})^2$ minus the variance penalty $\sigma_{t+1}^2$.

That way, a noisy TV will always be fitted with an higher variance that should have a similar order than the error of prediction, so both predictable and unpredictable transitions bonus will end up to zero (in average) after learning the model of dynamics of the whole environment.


**Summary Of The Review:**

(+) Nice and novel idea
(-) Too poor mathematical justifications and unclear description of the method.

I sincerely hope the paper will be improved for a further submission because I believe the idea has an high potential, that would be missed if it appears in its present slovenly form.

---

> ### Author Response · Authors · 2021-11-18
> **Response to QuEW**
>
> We appreciate the thoughtful review as well as the clear amount of effort taken to suggest ways to improve our manuscript. We respond
> to specific comments below with reviewer comments in bold and our responses below the bold text.
>
> **I am surprised that this idea was not explored in an earlier paper. Taking the prediction error minus the variance looks like the very first solution to try in order to prevent agents to stay watching a noisy TV. While my knowledge to this domain is limited, to my mind this is the first paper suggesting this idea.**
>
> Thank you, we believe that the simplicity of our idea is a positive.
>
> **Eq 3) is wrongly reported.**
>
> Thank you for catching this---you are correct---we will update this equation.
>
> **In Eq 5) $\mu_{s_{t+1}}$ should actually be $S_{t+1}$...**
>
> Yes, we agree. We will update this.
>
> **In Eq 6) $\hat{S}$ should be $\hat{\mu}$ for consistency**
>
> Sure, we also agree that this would make more sense.
>
> **Also I feel that there is a lot of paragraphes just to describe a very simple gaussian model..**
>
> Thanks for your comments on the number of paragraphs used to describe the loss function for aleatoric uncertainty estimation. We understand that this derivation may be simple but it is difficult to cater to multiple audiences---we have had other reviewers ask for more verbosity on how the loss function is derived!
>
> **Eq 5 is just the log of Eq 4 (with lambda = -1/pi). Then in Eq 7, it's just R = error - estimated variance. No need for $\eta$, the point is that these terms should have the same expectation at convergence of the fitted model with $\eta=1$.**
>
> $\eta$ and $\lambda$ are not required theoretically but we found that they can be useful hyperparameters when training AMA agents.
>
> **The bio-inspired aspect needs more content to be convincing:
> for example, reporting models or experimental results from the biological papers (for ex, the actual original curves that fig 5 is trying to reproduce).**
>
> We would like to compare to biological results but we are not aware of any experimental neuroscience literature which aims to disentangle the nature of uncertainty signalled by acetylcholine, hence why we develop the experimental blueprint and make predictions.
>
> Lastly, we are sincerely grateful for your concrete suggestions on how to update the method section, we will take these into account when writing our updated manuscript.

---

> > ### Comment · Reviewer_QuEW · 2021-11-22
> > **Thank for your response**
> >
> > Most of my concerns require minor changes in the equations and some rephrasing, and as the authors agree I believe they will apply these corrections.
> > also I've just noticed that in Pixel AMA, they use $\eta=2$, which confirms that in some practical cases it may produce better results than $\eta=1$, and justifies the existence of the parameter.
> >
> > Therefor I will increase my rating: the idea is worth a publication and the suggested corrections are doable.

---

> > > ### Author Response · Authors · 2021-11-24
> > > **Addressing the Method section**
> > >
> > > Thanks again for suggestions on updating the method section. We have implemented them and uploaded a revised manuscript. Please let us know if there are any further changes you would like us to make.

---

### Official Review · Reviewer_j6wR · 2021-11-03

**Correctness:** 2
**Technical Novelty And Significance:** 1
**Empirical Novelty And Significance:** 1
**Recommendation:** 3
**Confidence:** 4

**Main Review:**

The main weakness of the current work is that it misses relevant work which does the same thing while it seems to misinterpret certain parts of the literature. As an example, the authors state that "tractable epistemic uncertainty estimation with high dimensional data is an unsolved problem", citing Gal 2016 which demonstrates a method for estimating epistemic uncertainty in high dimensional data (that's MC Dropout). Also, there is a similar approach using Deep Ensembles to assess epistemic uncertainty. Although far from optimal, such approaches are already being used in epistemic uncertainty driven exploration. For example, in Planning to Explore ([https://arxiv.org/abs/2005.05960](https://arxiv.org/abs/2005.05960)) the authors use epistemic uncertainty estimation as an intrinsic reward. In fact, the authors in Related Work last paragraph, they cover a few efforts using epistemic uncertainty estimation as intrinsic rewards, while they don't mention what's the shortcomings of such work which they aim to solve with their proposal.

In terms of experimentation, the selected baselines are not sufficient to demonstrate how much better this method is. I'd expect to see a comparison with MC Dropout or Deep Ensembles based methods of assessing epistemic uncertainty (in the presence of heteroscedastic noise), or DUE ([https://arxiv.org/pdf/2102.11409.pdf](https://arxiv.org/pdf/2102.11409.pdf)).

**Detailed comments:**

C1: "this work presents aleatoric mapping agents which use single network deterministic uncertainty estimation" - is the reference correct there? Kendall & Gal work uses a Bayesian Neural Network, whereas deterministic uncertainty estimation was proposed here [https://openreview.net/forum?id=Fu7D6kQPzs4](https://openreview.net/forum?id=Fu7D6kQPzs4) (not cited in the text)

C2: For the epistemic and aleatoric uncertainty you cite Hullermeier & Waegeman. Can you also cite the earlier references? e.g. "Hora S (1996) Aleatory and epistemic uncertainty in probability elicitation with an example from hazardous waste management. Reliability Engineering and System Safety 54(2–3):217–223"

C3: "However, as far as we are aware, we are the first to compute aleatoric uncertainties with a scalable curiosity framework to reduce intrinsic rewards for those state transitions with aleatoric uncertainty" - I don't understand what is the novelty here. There is a lot of work disentangling epistemic from aleatoric uncertainty which works at scale. Can you please state as precise as possible what's the contribution in terms of the computation of aleatoric uncertainty?

C4: "We implicitly incentivise agents to seek epistemic uncertainties by removing the aleatoric component from the total prediction error" - If I'm not mistaken that's the common way to disentangle the epistemic from the aleatoric uncertainty which is usually derived by the law of total variance. What is the novelty here?

C5: "Possible because regularisation terms could absorb some epistemic uncertainty" - What does it mean to absorb some epistemic uncertainty? Epistemic uncertainty can be reduced or increased. Do you mean that through the regularization, epistemic uncertainty is reduced? If so, can you explain why this can happen?

C6: In the Minigrid experiments, can you try to implement the intrinsic reward based on the epistemic uncertainty as evaluated from ensemble Kendall & Gal 2017 models? That should be the Variance over expectations of the deep ensemble components $Var_{\theta \sim p(\theta \mid D)}[ \mu^\theta_{s_{t+1}} ]$.

C7: It's very difficult to follow the experimental protocol in section 5. How do you estimate the Acetylcholine in the simulations?

Finally, I'd like to suggest to the authors to try to use \citep instead of \cite whenever possible. Also try to use a more accessible colour for the citations as currently, it makes it challenging to read.

**Summary Of The Paper:**

This work proposes a method to disentangle epistemic from aleatoric uncertainty for avoiding the noisy TV problem which occurs when intrinsically motivated agents get rewarded for visiting states that have high irreducible uncertainty.

**Summary Of The Review:**

My recommendation is influenced by the following issues:
- The novelty of the contribution is not clear to me. I'd like to understand C3-C6 because overall it seems to me that epistemic uncertainty driven intrinsic rewards have been used before and here it's not compared thoroughly against such baselines (which are mentioned in related work)
- C7 - I don't understand how this section contributes towards understanding what type of uncertainty acetylcholine signals.

---

> ### Author Response · Authors · 2021-11-18
> **Response to j6wR Part 1**
>
> We thank the reviewer for taking the time to review our work and provide us with feedback, which we believe will lead to an improved manuscript. We address specific comments and questions below with the comments in bold and our responses to the comments under the bold text.
>
> **“The main weakness of the current work is that it misses relevant work which does the same thing while it seems to misinterpret certain parts of the literature. As an example, the authors state that "tractable epistemic uncertainty estimation with high dimensional data is an unsolved problem", citing Gal 2016 which demonstrates a method for estimating epistemic uncertainty in high dimensional data (that's MC Dropout).**
>
> We disagree that we are misinterpreting the literature by saying “tractable epistemic uncertainty is an unsolved problem”. We certainly do not think that after the invention of MC Dropout that tractable epistemic uncertainty estimation was solved. If it were solved then why would the papers mentioned later in the review (e.g. Ensembles and DUE) be gaining traction from the community? MC dropout produces worse uncertainty estimates than simple ensemble methods (e.g. Lakshminarayanan, Pritzel,  and Blundell (2017)). Furthermore, it is more computationally expensive in our reinforcement learning setting. Unlike in the supervised setting, using the epistemic uncerainty method from Kendall and Gal (2017) would require 50 forward passes through our prediction network at each training step because uncertainty would need to be computed online during training. Lastly, the utility of MC dropout for exploration in RL has not been demonstrated extensively. This is in contrast to random network distillation---a state of the art epistemic uncertainty driven intrinsic reward method that we compare to.
>
> **Also, there is a similar approach using Deep Ensembles to assess epistemic uncertainty. Although far from optimal, such approaches are already being used in epistemic uncertainty driven exploration.” For example, in Planning to Explore (https://arxiv.org/abs/2005.05960) the authors use epistemic uncertainty estimation as an intrinsic reward. In fact, the authors in Related Work last paragraph, they cover a few efforts using epistemic uncertainty estimation as intrinsic rewards, while they don't mention what's the shortcomings of such work which they aim to solve with their proposal.**
>
> The reviewer makes a good point here, we should have been clearer about the advantages that our approach affords over ensemble methods. While ensemble methods can be useful for exploration, they require considerably more more memory to store the ensemble network weights (usually around 5 networks). Besides the memory constraints of ensembles, they also require you train to multiple networks in parallel, which increases their computational expense. The AMA approach to exploration only requires one network (with an extra head for predicting uncertainties), reducing memory and computational complexity while also being simple to plugin to existing intrinsic reward methods.
>
> **In terms of experimentation, the selected baselines are not sufficient to demonstrate how much better this method is. I'd expect to see a comparison with MC Dropout or Deep Ensembles based methods of assessing epistemic uncertainty (in the presence of heteroscedastic noise), or DUE (https://arxiv.org/pdf/2102.11409.pdf).**
>
> We disagree that the baselines are not sufficient. RND is the backbone of SOTA atari RL agents (such as agent57) and it incentivises agents to search for epistemic uncertainty. In contrast, MC dropout has not been shown to be competitive for reinforcement learning exploration. We could compare to ensembles though we think that it is an unfair comparison as ensembles usually use 5 time the computational expense and memory of our method. This would make our retro game experiments very difficult with modest compute resources. We thank the reviewer for suggesting DUE. DUE is a very interesting work but its utility has only been demonstrated in a supervised learning setting. Instead of trying to adapt very recent epistemic uncertainty estimation methods for our baselines, we opted to compare to state of the art tried and true baselines.

---

> > ### Comment · Reviewer_j6wR · 2021-11-22
> > **Thank you for addressing the reviews**
> >
> > Thank you for addressing the reviews.
> >
> > What I meant by misinterpreting the literature is that you cite a paper that attempts to make the intractable inference problem to a tractable approximate inference problem after the sentence "... with high dimensional data is an unsolved problem". Although I agree with this sentence, I find that citing Gal 2016 after this makes the reader believe that the paper you cite states the same position, however, Gal 2016 offers a solution. I agree that Ensemble methods and RND can be superior methods to MC Dropout but my comment was about who you cite after this statement not whether MC Dropout has solved this problem - clearly, it hasn't and we agree on that.
> >
> > > "The novelty of our work is our demonstration that aleatoric uncertainty estimation is all that is required to compute intrinsic rewards that implicitly incentivise epistemic uncertainty."  and "Are you able to provide examples of work that only computes aleatoric uncertainty and uses that to explore in a curiosity-driven learning framework?"
> >
> > Yes, this to me seems very similar to this work and it's not covered in the related work section [https://arxiv.org/pdf/2102.08501.pdf](https://arxiv.org/pdf/2102.08501.pdf). From the abstract "Whereas previous work was focusing on model variance, we propose a principled approach for directly estimating epistemic uncertainty by learning to predict generalization error and subtracting an estimate of aleatoric uncertainty, i.e., intrinsic unpredictability ...  we illustrate its advantage against existing methods for uncertainty estimation on downstream tasks including sequential model optimization and reinforcement learning."
> >
> > Regarding comparison with ensembles. I agree with the authors that Ensembles are more expensive but the narrative of this paper is about making a better exploration method - the metric is the downstream task performance. The method you are suggesting here belongs to the single forward-pass uncertainty literature (e.g. DUE or [https://arxiv.org/pdf/2102.11582.pdf](https://arxiv.org/pdf/2102.11582.pdf)). I'd expect this to be part of the motivation story - then it will make sense to avoid comparing with ensembles since you have an extra assumption (single model).
> >
> > Regarding the benchmark. The author's argument is the following: RND is a good method but their method works better on some new benchmarks but they don't show how their method performs compared to RND on the original benchmark that RND was introduced. How can I know as a reviewer that this is not a benchmark lottery [https://arxiv.org/abs/2107.07002](https://arxiv.org/abs/2107.07002)? Then on Bank Heist, RND is dropped from the baselines and only IDF is used - Why? I'm not comfortable as a reviewer to suggest that the experimental section is supporting the main claims of the paper.
> >
> > Finally, regarding the ACh experiments. I agree with reviewer DE7e - this section shows that aleatoric uncertainty (also known as irreducible uncertainty) doesn't reduce over time whereas epistemic uncertainty (also know as reducible uncertainty) reduces. I can't see how this can prove or disprove any hypothesis about what kind of uncertainty ACh encodes.
> >
> > For the reasons above I will insist on my initial evaluation.

---

> > > ### Author Response · Authors · 2021-11-24
> > > **Thank you for engaging**
> > >
> > > Thank you for your engagement in our discussion and related work, which we genuinely appreciate.
> > >
> > > We cited Yarin Gal's PhD thesis for the sentence describing the difficulties of epistemic uncertainty prediction because we thought it would be a good place to point the interested reader for an introduction to the field and its challenges. However, we understand your point that this could be confusing so we will adjust this in future revisions.
> > >
> > > We are glad that we can agree that while useful, MC dropout leaves room for better approaches to epistemic uncertainty estimation.
> > >
> > > Thank you for pointing our attention to Direct Epistemic Uncertainty Prediction. This is an interesting work that we will definitely add to our related work section in future revisions. However, we do not believe it trumps the novelty of our approach for several reasons. Firstly, their paper appeared after ours on arxiv. Secondly, they learn to estimate generalisation error before subtracting aleatoric uncertainty from the estimated generalisation error, which is different to our approach. Thirdly, they have not employed their approach in a intrinsic motivation framework to avoid stochastic traps, which is the main motivation behind our work.
> > >
> > > If there are other works that you think we have missed please let us know---given that your main issue with our work is "misinterpreting literature" and "missing relevant works", we would appreciate any further examples so that we can address them in the future revisions.
> > >
> > > In terms of RND benchmarks, the benchmark environments we have chosen were due to their computational expense. We do not have the capacity to run 2 billion Atari frames for multiple seeds as applied in RND benchmarks. Instead, we opted for games that have been used previously in the curiosity literature that require less frames to train. Lastly, the purpose of the Bank Heist experiment was not to show that AMA curiosity is the state of the art on Bank Heist but rather to isolate a failure case of a particular flavor of curiosity and then show how our approach can help.
> > >
> > > The design of the proposed experiment aims to show that if ACh is measured in our proposed task and---for example---remains high in the face of aleatoric uncertainty, then we would have gathered evidence against the epistemic model of ACh. We have tried to make this clearer in our updated manuscript.
> > >
> > > Finally, we updated the citation style based on your suggestions. Please let us know if it is satisfactory.

---

> ### Author Response · Authors · 2021-11-18
> **Response to j6wR Part 2**
>
> **C1: "this work presents aleatoric mapping agents which use single network deterministic uncertainty estimation" - is the reference correct there? Kendall & Gal work uses a Bayesian Neural Network, whereas deterministic uncertainty estimation was proposed here https://openreview.net/forum?id=Fu7D6kQPzs4 (not cited in the text):**
>
> Kendall and Gal (2017) is the paper we cited in this section because it is their aleatoric uncertainty estimation method that we build upon. We understand they use a Bayesian neural network but this is not necessary for us because we are only computing aleatoric uncertainty.
>
> Nevertheless, we will add the DUE paper to our related work section.
>
> **C2: For the epistemic and aleatoric uncertainty you cite Hullermeier & Waegeman. Can you also cite the earlier references? e.g. "Hora S (1996) Aleatory and epistemic uncertainty in probability elicitation with an example from hazardous waste management. Reliability Engineering and System Safety 54(2–3):217–223"**
>
> Thank you for this suggestion, we will cite earlier works in our updated manuscript.
>
> **C3: "However, as far as we are aware, we are the first to compute aleatoric uncertainties with a scalable curiosity framework to reduce intrinsic rewards for those state transitions with aleatoric uncertainty" - I don't understand what is the novelty here. There is a lot of work disentangling epistemic from aleatoric uncertainty which works at scale. Can you please state as precise as possible what's the contribution in terms of the computation of aleatoric uncertainty?**
>
> The novelty of our work is our demonstration that aleatoric uncertainty estimation is all that is required to compute intrinsic rewards that implicitly incentivise epistemic uncertainty. This is useful because it is simple to implement and can be easily added onto existing intrinsic reward methods. Furthermore, the success of this method provides further evidence that aleatoric uncertainty could be represented by acetylcholine in the mammalian brain.
>
> **C4: "We implicitly incentivise agents to seek epistemic uncertainties by removing the aleatoric component from the total prediction error" - If I'm not mistaken that's the common way to disentangle the epistemic from the aleatoric uncertainty which is usually derived by the law of total variance. What is the novelty here?**
>
> The novelty of work is addressed in C3 above. Are you able to provide examples of work that only computes aleatoric uncertainty and uses that to explore in a curiosity driven learning framework?
>
> **C5: "Possible because regularisation terms could absorb some epistemic uncertainty" - What does it mean to absorb some epistemic uncertainty? Epistemic uncertainty can be reduced or increased. Do you mean that through the regularization, epistemic uncertainty is reduced? If so, can you explain why this can happen?**
>
> This point was merely a suggestion and is not central to the thesis of this work. The thinking was that the total prediction error in novel situations could be decreased with regularisation, therefore decreasing our (epistemic) intrinsic rewards in novel states. That is to say perhaps regularisation hurts performance when exploring because generalisability decreases intrinsic rewards in novel situations.
>
> **C6: In the Minigrid experiments, can you try to implement the intrinsic reward based on the epistemic uncertainty as evaluated from ensemble Kendall & Gal 2017 models? That should be the Variance over expectations of the deep ensemble components.**
>
> As stated above, we believe we have made sufficient comparisons to state of the art baselines. In the future we will consider implementing a comparison to deep ensembles.
>
> **C7: It's very difficult to follow the experimental protocol in section 5. How do you estimate the Acetylcholine in the simulations?**
>
> ACh is represented by ensemble variance in the epistemic case while ACh is represented by the predicted aleatoric uncertainty in the aleatoric model. We tried to make this clear in the manuscript but will try to improve on this in the future.
>
> Lastly, we can certainly change the citation style in the updated version.

---

### Author Response · Authors · 2021-11-23
**Update to Manuscript**

We have updated the manuscript to address the following concerns from reviewers:

- the method section has been reworked inline with QuEW's suggestions, with equations being written in more compact form and with what we hope is a clearer narrative

- the final section on the proposed mouse VR experiment has been edited with the aim of improving clarity

- we added citations to Hora S (1996) and DUE as suggested by j6wR

- we changed the citation style so that it is hopefully easier to parse as suggested by j6wR

In the coming days we will discuss the recent further comments from the reviewers.

---

### Decision · Program_Chairs · 2022-01-20

**Decision:**

Reject

**Comment:**

Authors present a method to disentangle epistemic from aleatoric uncertainty for avoiding the noisy TV problem during self-driven exploration. This is an important area where we need more ideas and experiments. The authors present a biologically inspired approach and through experiments. Although it doesn't present the state-of-the-art exploration in well-known RL environments, I acknowledge that new solutions to problems that were previously intractable often would face such an issue. The prediction to discriminate neuroscientific modulations that directly encode epistemic and aleatoric uncertainty is bold but not very specific. Unfortunately, as the reviewers noted, the manuscript in the current form doesn't quite meet the bar yet. I suggest comparing methods for directly estimating uncertainty. I also suggest adding discussion on the estimation bias for the epistemic uncertainty for the proposed method. I strongly encourage the authosr to continue this interesting line of work.